# Balneotherapy as a Complementary Intervention for Stress and Cortisol Reduction: Findings from a Randomized Controlled Trial

**DOI:** 10.3390/brainsci15020165

**Published:** 2025-02-07

**Authors:** Lolita Rapolienė, Dovydas Rapolis, Aelita Bredelytė, Giedrė Taletavičienė, Antonella Fioravanti, Arvydas Martinkėnas

**Affiliations:** 1Faculty of Health Sciences, Klaipėda University, 92294 Klaipeda, Lithuania; aelita.bredelyte@ku.lt (A.B.); arvydas.martinkenas@ku.lt (A.M.); 2Baltic Medics Clinic, 92334 Klaipeda, Lithuania; 3Faculty of Medicine, Vilnius University, 03101 Vilniaus, Lithuania; rapolis.dovydas@gmail.com; 4Druskininkai Hospital, 66251 Druskininkai, Lithuania; g.taletav@gmail.com; 5OMTh—Organizzazione Mondiale del Termalismo (World Hydrotermal Organization), 38056 Levico Terme, Italy; fioravanti7@virgilio.it

**Keywords:** balneotherapy, cortisol, distress, mineral water, peloids, salt therapy, well-being

## Abstract

Background: In our modern era, stress has become a pervasive challenge, affecting individuals across all ages and backgrounds. Acute or chronic stress and elevated cortisol levels are known to impair neurological function and hinder rehabilitation outcomes. Therefore, effective treatment methods that reduce stress, enhance mental health, and promote overall well-being are urgently needed. The aim of this study was to evaluate the seasonal impact of balneotherapy on distress, as measured by the General Symptoms Distress Scale (GSDS), and well-being, as assessed using the Arizona Integrative Outcomes Scale (AIOS), and the effect of winter balneotherapy on salivary cortisol levels. Methods: In 2023, a multicenter, single-blind, parallel-group, randomized controlled trial was carried out across six medical spa centers in Lithuania. Participants with a stress intensity greater than 3 points on the Visual Analogue Scale (VAS) underwent combined natural resource-based therapies over a 1- to 2-week treatment period. Outcomes were assessed using the General Symptom Distress and Arizona Integrative Outcomes scales, along with salivary cortisol measurements after winter intervention. Results: The results demonstrated a significant reduction in distress intensity by 1–3.5 points (VAS), with winter interventions showing greater efficacy compared to summer. Participants also experienced an increase in well-being by up to 3 points (VAS), improved stress management by up to 1.9 points (VAS), and a reduction in salivary cortisol levels by 0.9 units following winter-based treatments. Some gender differences emerged in specific groups. Conclusions: Our study provides robust evidence for the stress-reducing effects of balneotherapy, particularly highlighting the enhanced efficacy of winter interventions. These findings are especially relevant for neurological rehabilitation, where stress reduction and improved autonomic regulation can support neuroplasticity, recovery processes, and overall quality of life. This research offers valuable insights for developing holistic, seasonally optimized strategies to aid stress management and promote neurological health.

## 1. Introduction

In today’s modern world, stress has become an omnipresent adversary, affecting individuals of all ages and backgrounds. Stress is a psychological experience and physiological response mediated by a complex neuroendocrine, cellular, and molecular infrastructure within the nervous system [1]. Cortisol, the stress hormone, is released during stressful situations to help the body respond [2]. Cortisol regulates the stress response, metabolism, inflammation, blood pressure, and glucose availability and plays a role in the sleep–wake cycle. Far-reaching stress effects impact both mental well-being and physical health, ultimately compromising quality of life [3]. Extended exposure to stress can cause substantial psychological and pathological harm, especially in the brain areas responsible for cognition and emotional regulation, including the hippocampus, hypothalamus, amygdala, and prefrontal cortex [4]. These stress-induced changes can hinder recovery, exacerbate cognitive impairments, and accelerate neurodegeneration through mechanisms such as oxidative stress, apoptosis, and neuronal imbalance [5]. Stress has been found to exacerbate a range of neurological conditions, such as Alzheimer’s disease [6], stroke [7], traumatic brain injury [8], epilepsy [9], multiple sclerosis [10], and Parkinson’s disease [11]. Epidemiological studies also indicate a direct link between stress-related disorders and an increased risk of neurodegenerative diseases [12]. Stress is initially a natural response to adverse environmental factors; however, it is now understood that stress can be categorized into distinct types: distress, which serves as a non-specific contributor to disease, and eustress, which acts as a positive factor promoting health and longevity [13]. Unlike eustress, distress lasts longer, creates discomfort, overwhelms coping skills, and can cause anxiety and physical issues if unmanaged.

The importance of managing stress has never been more apparent, and stress management techniques, which encompass physical, emotional, and lifestyle control, have become essential in promoting overall well-being. Various relaxation techniques, including meditation, yoga, behavioral therapy, deep breathing, and massage, are effective in alleviating stress [14,15]. Among these, balneotherapy (BT) has gained increasing recognition for its therapeutic potential [16]. BT, which includes a variety of treatment modalities such as hydrotherapy, mineral baths, mud therapy, and the use of natural earth remedies, has been practiced for centuries, dating back to ancient civilizations that valued the healing properties of natural springs and mineral-rich waters [17]. Over time, BT has evolved into a structured treatment approach used in health resorts to promote physical and mental health. Research has demonstrated that balneotherapy is beneficial for improving overall health, reducing fatigue and stress, alleviating pain, and enhancing quality of life [18,19,20,21]. Additionally, different forms of water therapy have been associated with better sleep, reduced anxiety and depression, and improved health-related quality of life, particularly in older adults [22]. A systematic review has indicated that balneotherapy (BT) may impact cortisol levels in both healthy and ill individuals, enhancing stress resilience and providing significant benefits for managing stress-related conditions [23].

Despite historical and scientific evidence supporting the efficacy of mineral water and peloid treatments in stress reduction, comprehensive investigations into the broader therapeutic impacts remain limited. Understanding the connections between these natural therapies and stress regulation is crucial for advancing holistic health strategies. While previous studies have explored the relationship between BT and stress, our study introduces several novel aspects. First, it combines both objective and subjective measures to provide a holistic assessment of stress. Second, it offers a detailed analysis of specific stress-related symptoms. Third, it examines gender differences in stress and its predictive value. Finally, our study is the first to compare the seasonal effects of BT on stress and overall wellness. By addressing these gaps, our research provides new, clinically relevant insights into the impact of BT on stress regulation and well-being, contributing to a more comprehensive understanding of its therapeutic potential.

BT’s relevance to neurological rehabilitation is particularly significant due to its potential to improve autonomic nervous system regulation, reduce psychological stress, and induce physiological relaxation. These outcomes can positively influence neuroplasticity, support recovery processes, and enhance patients’ overall quality of life. For individuals suffering from conditions such as post-stroke fatigue, multiple sclerosis, Parkinson’s disease, and neuropathic pain—where stress and cortisol dysregulation play detrimental roles—BT could serve as a complementary intervention to optimize rehabilitation outcomes, but the more research in this field is needed.

The aim of this study was to evaluate the seasonal impact of balneotherapy on distress, as measured by the General Symptoms Distress Scale (GSDS), and well-being, assessed using the Arizona Integrative Outcomes Scale (AIOS), and the effect of winter balneotherapy on salivary cortisol levels.

## 2. Methodology

### 2.1. Study Design and Setting

This research was conducted as a multicenter, single-blind (researchers), interventional, randomized controlled study with parallel groups. It included two intervention periods during the winter (January–February) and summer (August–September) of 2023 across six medical spa centers in Lithuania: Egle and Draugystė in Druskininkai, Tulpė and Versmė in Birštonas, Gradiali in Palanga, and Atostogų parkas in the Kretinga region. The locations of the participating medical spas are provided in Figure A1.

The study adhered to the guidelines of the Declaration of Helsinki, received approval from the Kaunas Regional Research Ethics Committee (permission code BE-2-87, issued on 28 November 2022), and was registered on ClinicalTrials.gov (Identifier: NCT06018649, registered on 30 August 2023). All participants involved in the study provided written informed consent.

### 2.2. Study Participants and Sample Size

The inclusion criteria for the study were being between the ages of 18 and 65, experiencing moderate stress (greater than 3 points on a 10-point Visual Analogue Scale (VAS), and residing near one of the selected medical spas or being able to travel to the service centers. The exclusion criteria included uncontrolled or decompensated systemic diseases, active infections, malignant tumors, recent surgery or major trauma within the past year, use of balneotherapy within the previous 3 months, pregnancy or lactation, bleeding disorders, severe mental or physical health issues, and difficulties accessing the study locations. All participants provided written informed consent, which included detailed information about the study’s purpose, terms, and procedures before it began.

The sampling method used was a probabilistic nested (cluster) design, where each participant’s inclusion in the sample followed a multi-stage, criterion-based process. The required sample size for statistically significant comparisons of the rehabilitation effects on quantitative variables before and after the procedures was calculated using the G*Power Release 3.1.9.5 for Windows program, based on data from the authors’ previous published studies [16]. The smallest effect size of 0.32, which was statistically significant in assessing stress management in the control group, was used for the calculation. For effect sizes of 0.32, 0.4, and 0.5, the required group sizes were 79, 52, and 34 subjects, respectively. A rehabilitation effect size of 0.4 was selected, accounting for a small amount of data loss, so it was planned to have 55 participants in each group.

Individuals who met the eligibility criteria were assigned to the Klaipeda and Druskininkai clusters by trained administrative staff. They were then coded and randomly allocated to one of the six groups (1–6) by a statistician using a computer program after the initial screening (T0) at the study centers. Randomization was performed using a predefined SPSS method to ensure unbiased assignment to different treatment arms. The variables of age, sex, and baseline stress levels were analyzed using Pearson chi-square tests or ANOVA with Tukey’s post hoc test; no significant differences were found between the six groups.

The study groups were named using acronyms of treatment mode: 6ABT—6 days (1-week) ambulatory BT complex treatment, 11ABT—11 days (2 weeks) of ambulatory BT complex treatment, 11ABTNT—11 days (2 weeks) of ambulatory BT complex plus nature therapy treatment, 11SBT—11 days (2 weeks) of stacionary (inpatient) BT treatment in a medical spa, 11NT—11 days (2 weeks) nature therapy, 11C—11 days (2 weeks) control group without treatment. The groups of delayed procedures were named 6ABTS—6 days (1 week) of ambulatory BT complex treatment in summer and 11ABTS—11 days (2 weeks) of ambulatory BT complex treatment in summer. All group participants except 11BTS continue their daily work activity.

The periods of examination were T0—baseline/before treatment and T1—after treatment. A total of 1137 individuals were evaluated to determine their eligibility for participation in the study. Following the initial evaluation and reassessment of inclusion and exclusion criteria, final Excel files containing the allocation lists of eligible participants in Klaipėda (for the medical spas Gradiali and Atostogų Parkas) and Druskininkai (for the medical spas Eglė, Draugystė, Versmė, and Tulpė) clusters (N = 194 in Klaipėda and N = 179 in Druskininkai) were provided to the allocation manager responsible for randomization and allocation to treatment groups. Randomization was performed using the SPSS “Random Sample of Cases” function, which enables the selection of a subset of cases either by a specified percentage or an exact number. After the randomized allocation of participants into groups within the Klaipėda and Druskininkai clusters, they were assigned to specific medical spas. In the Druskininkai cluster:-The 6ABT group received BT treatments at Tulpė.-The 11ABT group received BT treatments at Versmė.-The 11ABTNT group received BT treatments at Draugystė.-Inpatient BT treatment was conducted at Eglė.

In the Klaipėda cluster:-The 6ABT and 11SBT groups were evenly distributed between Gradiali and Atostogų Parkas.-The 11ABT group received BT treatments at Atostogų Parkas.-The 11ABTNT group received BT treatments at Gradiali.

During the summer period, BT treatments in the Druskininkai cluster were equally distributed between Eglė and Draugystė, while in the Klaipėda cluster, they were equally distributed between Gradiali and Atostogų Parkas.

This article analyzes the short-term effects of BT across two intervention periods: winter and summer. The study flow diagram is presented in Figure A2. Six groups of participants (N = 340) received winter interventions, with assessments conducted after the treatment course, while two of the (nature therapy and control) groups (N = 59) received summer interventions (delayed), also with post-treatment assessments.

The assessment periods were as follows: winter-time groups (6ABT, 11ABT, 11ABTNT, 11SBT, 11NT, and 11C): T0: baseline/before treatment (27–29 January 2022) and T1: after treatment (6ABT group: 5 January 2023; remaining groups: 11–12 February 2023); summer-time groups (6ABTS, 11ABTS): T0: baseline/before treatment (11–13 August 2023) and T1: after treatment (6ABTS group: 21 August 2023; 11ABTS group: 28 August 2023). The researchers and physicians conducting participant screenings were blinded to group allocations. Data from 399 participants were included in the analysis.

### 2.3. Study Outcomes and Instruments

The primary outcomes of the study were the treatment’s effect on the intensity of distress symptoms, their management, and changes in salivary cortisol levels. The secondary outcome was the effect on integrative outcomes. Distress was assessed using the GSDS (General Symptoms Distress Scale, T. Badger), where participants rated the severity of 14 stress-related symptoms on a 10-point scale, with higher scores indicating greater distress. For stress management, participants rated their ability to manage these symptoms on a 10-point scale, with higher scores reflecting a better ability to cope with the symptoms [24]. The Cronbach’s alpha for the GSDS used was 0.771. For assessing the effect on integrative outcomes over the past 24 h, we used the AIOS scale (Arizona Integrative Outcomes Scale, Iris R Bell) [25]. This is a single-element visual analog scale where participants self-assess their overall feelings of spiritual, social, psychological, emotional, and physical well-being over the past 24 h or month. The AIOS can differentiate between individuals who are relatively ill and those who are relatively healthy. It also shows correlations with measures of distress, positive and negative effects, and indicators of positive mental states. Permissions to use the scales were obtained from the authors.

Laboratory outcomes—effect on salivatory cortisol level. The stress hormone cortisol was measured in saliva. Salivary cortisol measurement is a non-invasive and ecologically valid method for detecting early changes in brain health. It is also useful for evaluating the effectiveness of strategies aimed at relieving stress, improving brain health, and monitoring stress-related changes in the brain [26]. A specified amount of saliva was collected into the dispenser, and cortisol levels were tested in a certified laboratory in Germany. Elevated cortisol levels indicate higher stress. The method used for cortisol measurement was an enzyme-linked immunoassay (ELISA) with the Tecan Evolyzer device (Tecan, Männedorf, Switzerland). The test kit used was the Cortisol Saliva Elisa Tecan (REF: RE52611). The quantification sensitivity was 0.005 µg/dL (with a precision of 20%), with inter-assay variability ranging from 10.1% to 19.5% (CV: 13.2%) and intra-assay variability ranging from 3.2% to 6.1% (CV: 4.3%).

### 2.4. Treatments

All BT treatment groups (6ABT, 11ABT, 11ABTNT, 11SBT, 6ABTS, 11ABTS) followed the same BT regimen, which consisted of six days of daily treatment with one rest day in between. The regimen included 20 min of light exercises in a tap water pool, a 20 min mineral/geothermal water bath at 34–36 °C, 20 min of sapropel wrapping, and 25 min of salt therapy.

The nature therapy procedure involved a 45 min walk in nature (either in the forest or by the seaside), a series of simple low-intensity strength and breathing exercises, sensory experiences (such as aromatherapy from forest smells, listening to natural sounds, and collecting natural items), mindfulness therapy, and heliotherapy. The first session was guided by a physiotherapist, while the remaining sessions were carried out independently by the participants, with clear instructions provided and daily SMS reminders.

The mineral water used at the research centers was a highly mineralized sodium chloride (salt) solution, rich in calcium, magnesium, and sulfate, with a pH range of 5.71–7.54 and total dissolved solids (TDS) ranging from 16.750 to 82.445 g/L. The prescribed baths were of moderate mineralization to brine.

The peloids used for wrapping or bathing had a pH range of 6.6–7.0, humidity between 70.5% and 96%, and total mineralization ranging from 38.5 to 20,000 mg/L. These peloids varied in their composition, with the organic material content ranging from 14.32% to 91.96%, humic acid content between 1.22% and 28.25%, and fulvic acid content between 0.98% and 17.9%. Additionally, they contained minerals such as calcium, iron, magnesium, chlorine, sulfur, and silicon.

### 2.5. Statistical Analysis

Descriptive data were presented as means and standard deviations (SDs), with graphical representation of means and 95% confidence intervals (CIs). To compare continuous variables, independent 2-tailed *t*-tests were used, while chi-square tests were applied to compare the frequencies of variables within each rehabilitation group. A *z*-test was used to examine categorical variables across different categories. Analysis of variance (ANOVA) with Tukey HSD post hoc multiple comparison tests assessed the differences in the mean values of variables across study groups. Paired-sample *t*-tests were used to compare the means of variables at the start (T0) and end (T1) of treatment. If the normality assumption was not met, the Friedman non-parametric test was used to evaluate differences between variable values in the rehabilitation groups. Effect sizes, adjusted for sample size (Cohen’s *d* statistic), were also calculated. The effect size was calculated as Cohen’s *d*, with thresholds defined as 0.2 (small), 0.5 (medium), 0.8 (large), and 1.3 (very large). The regression analysis was conducted to identify predictive variables for the dependent variable GSDS_1 (distress symptoms intensity after treatment) and to determine potential prognostic parameters. The following measures were used as prognostic parameters for evaluating GSDS variable stress intensity: Work and Social Adaptation Scale [27]; Perceived Stress Scale [28]; Fatigue Assessment Scale [29]; State and Trait Anxiety Inventory (STAI-5) [30]; Depression Scale [31]; Sleep Quality Scale [32]; AIOS; Quality of Life (single question rated on a 5-point Likert scale); Pain (measured using the Visual Analogue Scale (VAS); and Blood Pressure (mmHg) and Heart Rate (beats/in) parameters. Permissions to use scales were obtained from the authors. Regression analysis was made using a linear regression model with the ENTER method applied. Pearson’s correlation coefficients were used to assess the relationship between saliva cortisol levels and GSDS variables. A *p*-value of <0.05 was considered statistically significant. All analyses were performed using SPSS (Statistical Package for the Social Sciences, Version 28.0, SPSS Inc., Chicago, IL, USA).

## 3. Study Results

### 3.1. The Characteristics of Study Participants

The characteristics of the study participants are presented in Table 1.

The groups did not differ in sociodemographic factors or post-COVID-19 parameters. However, they did differ in baseline stress intensity and stress management. Stress intensity was higher in the winter 6ABT group compared to the in 11NT (MD 1.8, *p* < 0.001) and in summer groups—6ATBS (MD 1.9, *p* = 0.002) and 11ABTS (MD 1.8, *p* = 0.009); in 11ABT and 11NT (MD 1.6, *p* = 0.009) and 6ABTS (MD 1.7, *p* = 0.011); in 11ABTNT than in 11NT (MD 1.8, *p* = 0.001) and summer groups—6ABTS (MD 1.9, *p* = 0.002) and 11ABTS (MD 1.6, *p* = 0.029). In terms of stress management, the 11NT group (MD −1.4, *p* = 0.027) and the 6ABTS group (MD -1.6, *p* = 0.031) showed better results compared to the 11SBT group.

Correlations between stress and age (−0.103, *p* = 0.043), working time (0.223, *p* < 0.001), physical activity (0.107, *p* = 0.036), COVID-19 infection (−0.108, *p* = 0.036), and stress management (−0.298, *p* < 0.001) were found.

### 3.2. The Treatment Effect on Distress

#### 3.2.1. The Effect on Distress Symptoms

The study results regarding changes in distress symptoms on the GSDS during winter and summer interventions are summarized in Table 2 and Table 3. Statistically significant improvements were observed in 11 (1-week BT) and 12 (2-weeks BT) out of 14 distress symptoms following winter interventions and in 3 (1-week BT) and 6 (2-weeks BT) symptoms after summer interventions.

Wintertime interventions
One-week BT interventions showed large effect sizes for fatigue, headache, anxiety, and memory/concentration; moderate effects for sleep, pain, depression, appetite, obstipation, and rash; and small effects for diarrhea. No significant effects were observed for nausea, vomiting, and tingling.Two-week outpatient BT interventions produced large effects for fatigue and anxiety; moderate effects for sleep, pain, headache, depression, memory/concentration, and tingling; and small effects for appetite, nausea, obstipation, and rash. No significant effects were observed for vomiting and diarrhea.BT combined with nature therapy demonstrated large effects on fatigue, anxiety, and memory/concentration; moderate effects for sleep, appetite, and nausea; and small effects for pain, depression, vomiting, obstipation, diarrhea, tingling, and rash. No significant effects were observed for pain and headache.Inpatient BT interventions had very large effects on fatigue, large effects for pain, anxiety, and tingling; moderate effects for sleep, memory/concentration, appetite, and nausea; and small effects for headache, depression, obstipation, and diarrhea. No significant effects were observed for vomiting and rash.Nature therapy alone significantly reduced pain and headache with moderate and small effects, respectively, but slightly worsened diarrhea.Minimal effects were observed in the control group, including small reductions in fatigue and tingling, alongside a slight worsening of obstipation.

Summertime interventions Significant positive results after the 1-week summer BT were observed in fatigue and rash (moderate effect size) and anxiety (small effect size).The 2-week summer BT demonstrated more extensive positive changes, including a reduction in fatigue (large effect size), pain, anxiety, and memory and concentration problems (moderate effect size), as well as tingling and sleep problems (small effect size) (Table 3).

After conducting post hoc tests for multiple comparisons, significant differences were identified between groups before and after treatment (Table 3). At baseline, all distress symptoms, except for diarrhea, were more pronounced during the winter season, making it challenging to accurately assess differences following the treatment course.

There were no significant differences observed in the summer groups when comparing the data before and after treatment. However, in the winter season groups, the effect on fatigue significantly differed between treatment modalities after treatment. The inpatient group showed superior outcomes compared to the outpatient group (MD −1.36, *p* = 0.02), BT with nature therapy (MD −1.25, *p* = 0.046), and the nature-only (MD −2.21, *p* < 0.001) and control groups (MD −2.30, *p* < 0.001). Memory/concentration showed greater improvement in the 1-week BT group compared to the inpatient group (MD −1.26, *p* = 0.008).

In reducing diarrhea, BT combined with nature therapy was more effective than the 2-week BT group (MD −0.57, *p* = 0.016). Tingling and numbness were significantly reduced by inpatient BT procedures compared to outpatient treatments (MD 0.83, *p* = 0.036). Additional significant differences were identified between treatment groups when compared to the nature-only and control groups (Table 3).

#### 3.2.2. The Effect on Distress Symptom Intensity

Figure 1 presents the data on overall distress symptom intensity changes across treatment groups. Statistically significant reductions in distress intensity were observed in both the winter and summer BT groups between the baseline and post-treatment periods.

In the 1-week winter ambulatory BT group, stress intensity decreased by 3.5 points (VAS) (*p* < 0.001, Cohen’s *d* = 1.1, large effect).Similarly, reductions of 2.6 points (VAS) (*p* < 0.001, Cohen’s *d* = 1.0, large effect) and 2.9 points (VAS) (*p* < 0.001, Cohen’s *d* = 0.8, large effect) were observed in the 2-week winter ambulatory BT group and the 2-week BT with nature therapy group, respectively.In the 2-week inpatient BT group, distress intensity was reduced by 2.6 points (VAS) (*p* < 0.001, Cohen’s *d* = 1.3, very large effect).During the summer, stress intensity decreased by 1.0 point (VAS) (*p* = 0.032, Cohen’s *d* = 0.4, small effect) in the 1-week ambulatory BT group and by 1.8 points (VAS) (*p* < 0.001, Cohen’s *d* = 0.8, large effect) in the 2-week ambulatory BT group.Notably, a reduction was also observed in the control group, with stress intensity decreasing by 1.2 points (VAS) (*p* < 0.001, Cohen’s *d* = 0.6, moderate effect).No significant changes in distress intensity were found in the nature therapy group.The between-group comparison revealed significant differences in baseline distress intensity (*p* < 0.001, ANOVA effect size = 0.1) and post-treatment distress intensity (*p* = 0.031, ANOVA effect size = 0.04) (Table 1). After treatment, a significant difference was noted between the 1-week BT group and the control group, with a mean difference (MD) of −1.2 (*p* = 0.045).

#### 3.2.3. The Effect on Distress Symptom Management

Significant improvements in distress management were observed in three winter intervention groups following treatment (Figure 2).

The 2-week ambulatory BT group demonstrated an improvement of 1.1 points (*p* = 0.002, Cohen’s *d* = 0.4, small effect), the 2-week BT plus NT group showed an improvement of 1.0 points (*p* = 0.012, Cohen’s *d* = 0.3, small effect), and the 2-week inpatient BT group exhibited the largest improvement, with a 1.9-point increase (*p* < 0.001, Cohen’s *d* = 0.9, large effect). The control group experienced non-significant reductions in stress management, while non-significant positive changes were observed in the short-term winter and summer intervention groups.

Between-group comparisons showed significant differences in baseline stress intensity (*p* < 0.001, ANOVA effect size 0.1) (Table 1). However, no significant differences were found between the groups after treatment (*p* = 0.181).

### 3.3. The Treatment Effect on Salivary Cortisol

The changes in salivary cortisol levels following winter interventions are illustrated in Figure 3. Salivary cortisol levels significantly decreased after BT treatments: a reduction of 0.67 nmol/L was observed in the 1-week BT group (*p* = 0.013, Cohen’s *d* = 0.3, small effect), a reduction of 0.69 nmol/L was observed in the 2-week ambulatory BT group (*p* = 0.014, Cohen’s *d* = 0.4, small effect), a reduction of 0.87 nmol/L was observed in the BT plus nature therapy (NT) group (*p* = 0.006, Cohen’s *d* = 0.4, small effect), and a reduction of 0.74 nmol/L was observed in the inpatient group (*p* = 0.034, Cohen’s *d* = 0.2, small effect).

Between-group comparisons indicated significant differences in baseline cortisol levels (*p* = 0.040, ANOVA effect size = 0.04). However, no significant differences in cortisol levels were observed between the groups following treatment (*p* = 0.283).

### 3.4. The Treatment Effect on Integrative Outcomes

The changes in integrative outcomes, as measured by the sense of overall well-being, are depicted in Figure 4. Winter interventions significantly improved the sense of well-being. Specifically, 1-week ambulatory BT increased well-being by 2.7 points (*p* < 0.001, Cohen’s *d* = −1.1, large effect), 2-week ambulatory BT increased well-being by 1.9 points (*p* < 0.001, Cohen’s *d* = −0.8, large effect); the BT plus NT group increased well-being by 3.0 points (*p* < 0.001, Cohen’s *d* = −1.4, very large effect); and inpatient BT increased well-being by 1.4 points (*p* < 0.001, Cohen’s *d* = −0.8, large effect).

Between-group comparisons of integrative outcomes showed significant differences in baseline integrative outcomes (*p* < 0.001, ANOVA effect size = 0.15). At baseline, the nature therapy, control, and summer intervention groups had higher scores (indicating a better sense of well-being) compared to the winter interventions. After treatment, significant differences emerged between groups (*p* < 0.001, ANOVA effect size = −0.16).

The 1-week winter ambulatory BT treatment (6ABT group) led to a greater improvement compared to the inpatient treatment (11SBT group) (MD = 1.2, *p* = 0.003), nature therapy (11NT group) (MD = 1.2, *p* = 0.008), and the control treatment (MD = 1.4, *p* < 0.001).The 2-week winter ambulatory BT treatment (11ABT group) had a greater effect compared to the summer treatment: 1-week BT (6ABTS group) (MD = −1.2, *p* = 0.026) and 2-week BT (11ABTS group) (MD = −1.5, *p* = 0.003).The BT combined with nature therapy group (11ABTNT group) showed a greater improvement compared to the inpatient (MD = 1.1, *p* = 0.003), nature therapy (MD = 1.2, *p* = 0.008), and the control groups (MD = 1.4, *p* < 0.001).The inpatient BT group (11SBT group) demonstrated greater improvement compared to both summer groups: 1-week (MD = −1.7, *p* < 0.001) and 2-week (MD = −1.9, *p* < 0.001).

### 3.5. Analysis of Stress Outcomes for Predictive Value and Gender Differences

We conducted a regression analysis with the dependent variable GSDS_1 (distress symptoms intensity after treatment) to identify potential prognostic parameters (Table 4).

The model demonstrated a coefficient of determination (R^2^) of 0.362, with a statistically significant regression model (*p* < 0.001, ANOVA). The analysis identified significant beta coefficients for the following predictors: work and social adaptation (*β* = 0.32), state anxiety (*β* = 0.33), trait anxiety (*β* = −0.29), and integrative outcomes (*β* = −0.26). Notably, saliva cortisol levels and other variables did not demonstrate significant predictive value. We attempted to assess the correlation between subjective stress intensity (as measured by the GSDS item) and saliva cortisol levels using Pearson correlation analysis, but the correlation was not significant (pre-treatment: r = 0.027, *p* = 0.320; post-treatment: r = 0.041, *p* = 0.244).

An increase in the predictor variables “work and social adaptation” and “state anxiety” was associated with an increase in the dependent variable distress symptom intensity, while the “sense of well-being/integrative outcomes” and “trait anxiety” were associated with a decrease in distress intensity. These results are summarized in Table 4.

Since salivary cortisol did not demonstrate significant predictive value for the GSDS variable of stress intensity (Table 4), we performed a regression analysis with salivary cortisol as the dependent variable and all items of the GSDS. The prognostic value of cortisol was independent of GSDS symptoms at both the beginning and end of rehabilitation. None of the models reached statistical significance (ANOVA *p* > 0.05). Pearson correlation analysis between salivary cortisol and all GSDS scale variables revealed some correlations. After treatment, a significant correlation was found between salivary cortisol and appetite loss (r = 0.106, *p* = 0.037). Additionally, weak or marginally significant correlations were observed for pain (r = 0.094, *p* = 0.057), anxiety (r = 0.080, *p* = 0.089), and vomiting (r = 0.079, *p* = 0.091). No other significant correlations were identified.

In the overall winter and summer intervention sample, no significant differences in stress intensity were found between males and females at the beginning and end of treatment. Before treatment (T0), the average stress intensity for males was 5.44 (SD 2.40), while for females it was 5.67 (SD 2.25) (t = −0.811, df = 392, *p* = 0.418). After treatment (T1), the average stress intensity for males decreased to 3,88 (SD 2.04) and for females it decreased to 3.45 (SD 1.99) (t = 0.122, df = 275, *p* = 0.903). Similar results were found in the winter interventions sample: before treatment (T0), the average stress intensity for males was 5.65 (SD = 2.40), while for females it was 5.91 (SD = 2.16) (t = −0.911; df = 319; *p* = 0.363); after treatment (T1), the average stress intensity for males decreased to 3.32 (SD = 1.99) and for females it decreased to 3.55 (SD = 2.00) (t = −0.851; df = 317; *p* = 0.396).

When analyzed by winter groups (Table 5), before treatment (T0), males had lower stress than females, with a significant difference in the 1-week BT group (6ABT) (*p* = 0.024, effect size = 0.72). The other groups showed no significant gender differences. After treatment (T1), stress levels decreased for both genders, but the 1-week BT group still showed a significant gender gap (*p* = 0.002, effect size 0.79), with males benefiting more. Overall, gender differences in stress remained only in the 1-week BT group, while the other groups showed balanced reductions over time.

When evaluating summer stress intensity differences by gender, the following results were found. Before treatment, there was no significant change in stress intensity for males (t = −0.177; *p* = 0.860) or females (t = −0.305; *p* = 0.762). After treatment, there was no significant change in stress intensity for males (t = 0.839; *p* = 0.409) but a significant reduction in stress intensity for females (t = 2.975; *p* = 0.006, large effect) (Table 6).

Before treatment, cortisol levels differed significantly between men and women (the mean cortisol level for men was 3.62 nmol/L, which was higher than that for women at 2.98, *p* = 0.036). After treatment, this difference became statistically insignificant (*p* = 0.22). When analyzing gender differences in the winter intervention groups, several significant differences were observed (Table 7). Before treatment (T0), males had significantly higher cortisol levels than females in the 1-week (6ABT) group (*p* = 0.047, effect size = 0.78). The other groups showed no significant gender differences (*p* > 0.05), although males generally had slightly higher cortisol levels than females. After treatment (T1), cortisol levels decreased in both genders. In the 2-week BT with nature therapy group (11ABTNT), males still had significantly higher cortisol than females (*p* = 0.043, effect size = 0.55), but the reduction in cortisol was greater in males (0.96 vs. 0.76 nmol/L). No significant gender differences remained in the other groups (*p* > 0.05).

To determine the effects of BT on different genders, a paired-samples *t*-test was used to assess the change in salivary cortisol after treatment. The results were significant for both genders. For men, the mean difference (MD) was 0.84 (SD = 0.26), with a 95% confidence interval (CI) of 0.3275 to 1.355, *p* < 0.002, and an effect size of 0.39. For women, the mean difference (MD) was 0.52 (SD = 2.42), with a 95% confidence interval (CI) of 0.2030 to 0.8401, *p* = 0.001, and an effect size of 0.22. According to the effect size, the treatment had a larger effect in men.

## 4. Discussion

This study offers an in-depth analysis of the effectiveness of balneotherapy and its variations, including combinations with nature therapy, in alleviating distress symptoms, altering salivary cortisol levels, and enhancing an overall sense of well-being. We found that the intensity of distress was reduced using different modes of procedures that include different natural resources, regardless of treatment duration, composition, and season (MD 1–3.5 points, VAS). The 2-week interventions, both ambulatory and inpatient, consistently produced superior results compared to shorter 1-week treatments.

The findings suggest that winter interventions were more effective than summer interventions at reducing distress symptoms and enhancing well-being, with winter treatments improving twice as many symptoms as the 2-week summer intervention. The 2-week inpatient BT program showed the greatest benefits, particularly for fatigue, pain, anxiety, and tingling, with additional improvements seen when combined with nature therapy. According to research, the integration of BT with green exercises further enhanced quality of life and emotional well-being, especially for patients with lower back pain [33]. Although the control group showed the smallest reduction in stress intensity compared to the winter intervention groups, the statistically significant decrease may be influenced by uncontrolled variables, such as natural fluctuations, seasonal changes, or lifestyle modifications. Participants may have gained a better understanding of their feelings and benefited from reflection and attention from medical staff during the examination. These factors, even without active treatment, could have contributed to the reduction in stress levels. This highlights the need to consider such influences when interpreting the results.

In contrast, summer BT interventions showed only modest improvements, mainly in fatigue, rash, and anxiety. This seasonal variation suggests that winter treatments provide a more controlled and intensive therapeutic environment. Between-group analysis found that inpatient treatment was most effective for fatigue and tingling, while short-term winter BT was better for memory and concentration. No significant differences were observed among summer groups.

Despite some treatment-specific effects, baseline differences between winter and summer groups make it difficult to establish a direct causal link. Interestingly, memory and concentration improved more in the 1-week BT group than in the inpatient group, possibly due to the stimulating effects of returning to work. In contrast, inpatient participants, removed from work and deeply relaxed, may require more time to regain focus. The interpretation of these results remains complex and may be influenced by the number of statistical comparisons performed, warranting further research.

Saliva cortisol levels, an objective marker of stress, were significantly reduced following winter BT interventions. However, no significant differences were observed between groups after treatment, indicating that while BT is effective in lowering cortisol, the effects may not vary drastically across different modalities. This finding supports BT as a viable intervention for stress reduction and highlights its potential integration into broader stress management and neurological rehabilitation strategies.

The sense of overall well-being improved significantly across all winter BT interventions, with very large effects observed in the BT plus NT group. Improvements in well-being were less pronounced in summer interventions and the control group, emphasizing the enhanced efficacy of winter interventions. Notably, the regression analysis indicated that a higher sense of well-being was associated with a decrease in distress intensity, underscoring the importance of integrative outcomes in evaluating treatment success.

By comparing our results with existing trials and the literature, we can elucidate the unique contributions and implications of mineral water and peloid treatments in stress reduction. Firstly, our study corroborates previous research demonstrating the stress-alleviating effects of balneotherapy. It has been demonstrated that BT provides a variety of benefits, especially in improving quality of life, overall well-being, and physical fitness [34]. Research indicates that even short-term BT can reduce subjective stress [20] and lower cortisol levels in saliva [21,35]. Balneotherapy, which includes spa therapy and aquatic exercises, helps reduce mental stress and sleep disturbances, while also enhancing overall health in individuals with suboptimal health [36]. A study comparing the physical and mental effects of a 2-week bathing versus showering intervention found that bathing was more beneficial than showering. VAS scores showed significantly greater improvements in fatigue, stress (−15.3 vs. −10.2), pain, and self-reported health (9.9 vs. 5.5) following the bathing intervention compared to the showering intervention [21]. From a physiological standpoint, BT treatment has the potential to boost serum β-endorphin levels and may influence cortisol levels, enhancing an individual’s resilience to stress without disrupting the hormone’s natural circadian rhythm [37,38]. A significant effect has been observed on serotonin, a neurotransmitter that influences various biological and behavioral processes, including sleep, food intake, circadian rhythms, pain perception, cognition, reproductive activity, mood, and responses to anxiety or stress [39]. Research revealed positive effects of BT in reducing stress, fatigue, mood disturbances, feelings of depression and burnout, and waist circumference [36] as well as improving quality of life, sleep, psychomotor performance, and mental activity [16,23,40]. Compared to progressive muscle relaxation, BT provides greater relaxation and similarly reduces cortisol levels in saliva [41]. The study conducted by Toda et al. demonstrated that salivary cortisol levels decreased after spa bathing, based on samples collected from 12 healthy males before and after the treatment [42]. A new study shows that daily bathing in a neutral bicarbonate ionized bath can reduce psychological tension, improve sleep quality, enhance immune system function, and have a positive impact on health for those experiencing daily life stressors [43]. The systematic review concluded that spa therapy, whether combined with peloid therapy or not, effectively reduced cortisol levels in both suboptimal-health and ill individuals. Cortisol levels decreased after a single session and showed further improvement following a complete cycle of spa therapy [23]. The 2017 study involving 4265 hot spring users demonstrated that hot spring bathing led to significant improvements in insomnia, anxiety, and depression [20]. Various scientific evidence confirms that BT is an effective and valuable supplementary method for reducing stress and enhancing mental health [44]. Our findings align with these studies, providing further evidence of the efficacy of balneotherapy as a non-pharmacological approach to stress management in a broader context.

In our study, the significant predictors of perceived stress highlight the essential role of psychological and social factors in the regulation of distress, with biological markers such as cortisol appearing to have a less direct impact. The regression analysis revealed that work and social adaptation, state anxiety, trait anxiety, and integrative outcomes were key predictors of distress symptom intensity. Specifically, higher levels of poor work/social adaptation and state anxiety were associated with greater distress, while better integrative outcomes and lower trait anxiety were linked to reduced distress.

Interestingly, salivary cortisol did not demonstrate significant predictive value for distress intensity. Although weak correlations were found between cortisol levels and appetite loss (r = 0.106, *p* = 0.037), as well as marginal correlations with pain, anxiety, and vomiting, the overall association between cortisol and distress intensity remained limited. These results suggest that cortisol may not serve as a reliable or direct predictor of distress intensity.

This raises the possibility that cortisol’s role in stress measurement may be more nuanced or that other subjective stress scales might offer a better means of assessing the complex relationship between physiological and psychological stress. Future research should explore cortisol’s relationship with other stress measurement tools to clarify its validity as an objective biomarker of stress and to determine whether it can more effectively contribute to stress assessments in clinical settings.

Overall, while no significant gender differences in stress intensity were found in the full winter and summer sample, differences emerged in specific groups. Subjective stress intensity showed greater reductions in females after summer treatment, whereas in winter, males in the 1-week BT group benefited more, although significant gender differences were observed in this group, where males had lower stress levels than females. In contrast, objective measures of stress (salivary cortisol) were initially higher in males but equalized after treatment, except in the 2-week BT with nature therapy group, where males maintained higher cortisol levels despite a greater reduction in them. These findings highlight variations in how men and women respond to stress interventions, with subjective and objective parameters showing different patterns. The treatment effects on cortisol were significant for both genders, with a larger effect size in males. Our finding of higher perceived stress in women supports research showing well-established sex differences in hypothalamic–pituitary–adrenal axis responses, with women typically exhibiting stronger reactions to stress [45]. Other researchers have found significant improvements in physical and social functioning, pain, and vitality in BT-treated patients, regardless of age or gender [46].

The study results highlight significant seasonal and treatment-related differences, emphasizing the potential of BT in the context of brain health and neurological rehabilitation. Stress negatively impacts brain injuries and diseases, affecting brain regions through various pathways. It can lead to mental disorders in children and adults, impairing neural plasticity and increasing the risk of emotional issues [4]. Chronic stress and associated distress symptoms are known to negatively affect cognitive function, emotional regulation, and overall neurological health. The observed improvements in memory, concentration, and anxiety suggest that BT can support cognitive rehabilitation and mental well-being in patients with neurological impairments. The large effect sizes for fatigue and pain also indicate its utility in managing common neurological symptoms, such as chronic fatigue syndrome and neuropathic pain. The demonstrated efficacy of BT, particularly in winter, highlights its potential as a non-invasive, holistic intervention for stress-related neurological conditions. The findings support the integration of BT into multidisciplinary treatment frameworks for neurological rehabilitation. Its stress-reducing effects, evidenced by cortisol reductions and improvements in stress management, align well with therapeutic goals in managing neurodegenerative diseases, stroke recovery, and traumatic brain injury. Furthermore, the inclusion of nature therapy as an adjunctive therapy offers a promising avenue for enhancing the therapeutic impact of BT, particularly for cognitive and emotional outcomes.

### Limitations and Future Directions

While the study has notable strengths, it also has limitations, such as the relatively small sample size in certain groups, particularly during the summer season, the lack of salivary cortisol results for the summer group, the complex nature of the spa effect, and the potential for confounding variables. There was also a lack of standardization for the content of natural resources, application mode used, and longer or shorter rest periods during activities. These factors make it challenging to compare results across groups and draw definitive conclusions. Standardized protocols would enhance the comparability of studies. The seasonal differences in baseline distress symptoms, particularly the higher levels observed during winter, may have influenced treatment outcomes. The lack of significant changes in some variables, such as saliva cortisol and distress symptoms in summer interventions, warrants further investigation. Future research should investigate the long-term effects of BT and its potential application to various populations, including individuals with specific neurological disorders, and the use of objective together with subjective to measure the outcomes. The potential role of NT as a standalone or complementary therapy also merits further exploration to optimize treatment protocols for brain health and neurological rehabilitation. Further research on stress mechanisms and therapeutic approaches is needed.

## 5. Conclusions

Our study highlights the significant stress-reducing benefits of combining natural resources, particularly balneotherapy and nature therapy, with the enhanced efficacy observed during winter and inpatient treatment modes.

This study underscores the efficacy of BT, particularly during winter, in reducing distress symptoms, improving stress management, and enhancing overall well-being. The findings highlight the potential of BT as an integrative approach in brain health and neurological rehabilitation. By addressing both psychological and physiological dimensions of stress, BT offers a holistic, innovative, and evidence-based intervention for improving quality of life and supporting neurological recovery.

## Figures and Tables

**Figure 1 brainsci-15-00165-f001:**
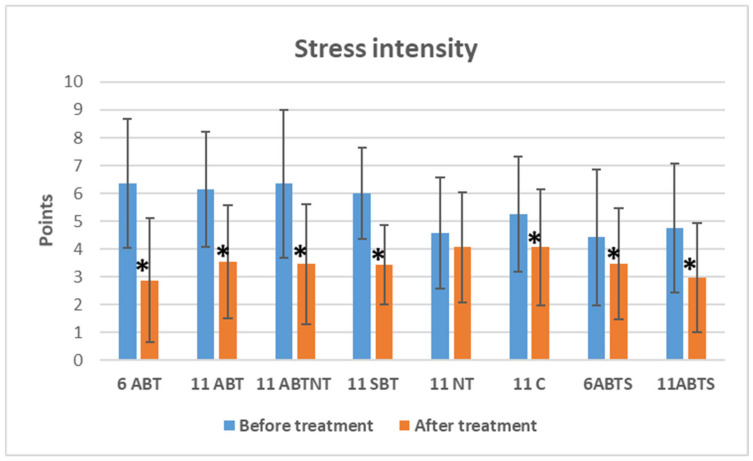
The change in distress intensity in study groups after interventions. * significant changes, *p*-value < 0.05. Abbreviations: 6ABT—1-week of ambulatory BT complex treatment, 11ABT—2 weeks of ambulatory BT complex treatment, 11ABTNT—2 weeks of ambulatory BT complex plus nature therapy treatment, 11SBT2 weeks of inpatient BT complex treatment, 11NT—2-weeks of nature therapy, 11C—2 weeks control group. 6ABTS—1 week of ambulatory BT complex treatment in summer, 11ABTS—2 weeks of ambulatory BT complex treatment in summer.

**Figure 2 brainsci-15-00165-f002:**
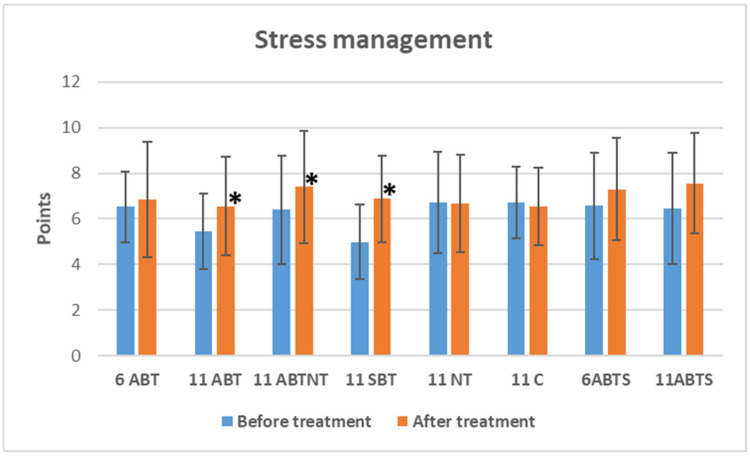
The change in distress symptom management in study groups after interventions. * significant changes, *p*-value < 0.05. Abbreviations: 6ABT—1-week of ambulatory BT complex treatment, 11ABT—2 weeks of ambulatory BT complex treatment, 11ABTNT—2 weeks of ambulatory BT complex plus nature therapy treatment, 11SBT2 weeks of inpatient BT complex treatment, 11NT—2-weeks of nature therapy, 11C—2 weeks control group, 6ABTS—1 week of ambulatory BT complex treatment in summer, 11ABTS—2 weeks of ambulatory BT complex treatment in summer.

**Figure 3 brainsci-15-00165-f003:**
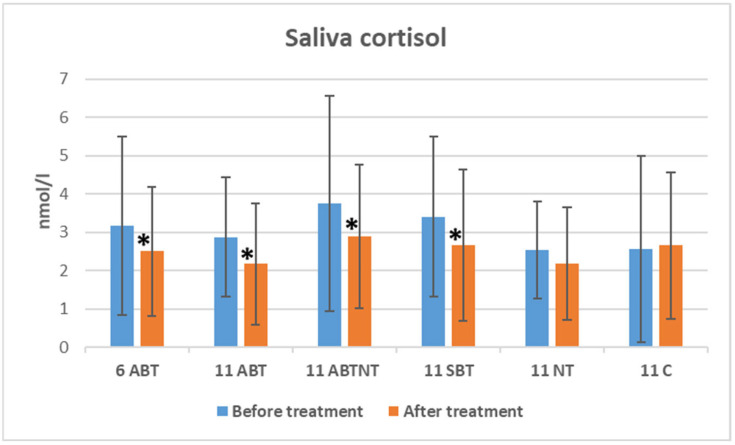
The change in saliva cortisol in study groups after winter interventions. * significant changes, *p*-value < 0.05. Abbreviations: 6ABT—1-week of ambulatory BT complex treatment, 11ABT—2 weeks of ambulatory BT complex treatment, 11ABTNT—2 weeks of ambulatory BT complex plus nature therapy treatment, 11SBT2 weeks of inpatient BT complex treatment, 11NT—2-weeks of nature therapy, 11C—2 weeks control group.

**Figure 4 brainsci-15-00165-f004:**
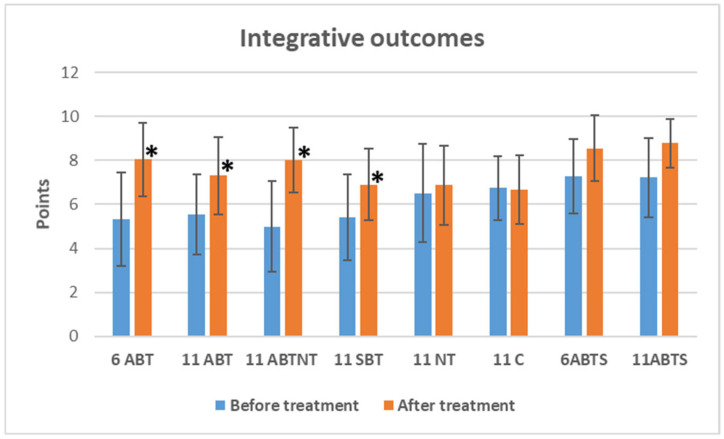
The change in integrative outcomes in study groups after interventions. * significant changes, *p*-value < 0.05.Abbreviations: 6ABT—1-week of ambulatory BT complex treatment, 11ABT—2 weeks of ambulatory BT complex treatment, 11ABTNT—2 weeks of ambulatory BT complex plus nature therapy treatment, 11SBT2 weeks of inpatient BT complex treatment, 11NT—2-weeks of nature therapy, 11C—2 weeks control group, 6ABTS—1 week of ambulatory BT complex treatment in summer, 11ABTS—2 weeks of ambulatory BT complex treatment in summer.

**Table 1 brainsci-15-00165-t001:** Participant characteristics in the study.

Parameters	6ABT N = 59	11ABT N = 63	11ABTNT N = 63	11SBT N = 61	11NT N = 43	11C N = 51	6ABTS N = 30	11ABTS N = 29	χ^2^, *p*-Value
Age, years ± SD	45.3 ± 10.0	49.0 ± 11.5	46.3 ± 10.1	49.0 ± 10.5	44.8 ± 12.8	46.0 ± 10.9	45.5 ± 12.1	48.3 ± 10.6	0.279 ^a^
BMI ^a^ ± SD	26.6 ± 5.3	26.4 ± 4.9	2673 ± 5.5	27.1 ± 4.5	25.6 ± 4.5	27.0 ± 5.5	24.7 ± 3.6	27.4 ± 5.6	0.447
Gender, Female (%)	44 (75.9)	47 (87)	41 (66.1)	44 (72.1)	34 (79.1)	37 (77.1)	26 (86.7)	21 (75)	0.108
Marital status, Married N (%)	45 (77.6)	35 (64.8)	42 (68.9)	35 (60.3)	26 (65.0)	36 (75.0)	19 (65.5)	22 (78.6)	0.355
Education, University level N (%)	39 (67.2)	33 (61.1)	29 (47.5)	37 (62.7)	31 (75.6)	32 (66.7)	23 (76.7)	18 (64.3)	0.085
Profession, White-collar N (%)	29 (50.0)	25 (47.2)	27 (44.3)	35 (59.3)	23 (56.1)	21 (43.8)	18 (62.1)	12 (42.9)	0.523
Nature of work, Sedentary N (%)	31 (53.4)	22 (40.7)	28 (48.3)	29 (50.9)	22 (53.7)	18 (38.3)	17 (58.6)	11 (39.3)	0.946
Work experience, >20 years N (%)	33 (56.9)	27 (50.0)	34 (55.7)	40 (67.8)	24 (60.0)	25 (52.1)	18 (62.1)	21 (75.0)	0.169
Working time, ≤8 h N (%)	29 (50.0)	32 (59.3)	30 (50.0)	24 (43.6)	20 (52.6)	22(45.8)	15 (53.6)	13 (46.4)	0.989
Rest time,7–8 h N (%)	24 (41.4)	24 (45.3)	26 (43.3)	23 (39.0)	21 (52.5)	23 (50.0)	18 (62.1)	11 (40.7.4)	0.797
COVID-19 infection N (%)	39 (67.2)	31 (57.4)	41 (66.1)	32 (54.0)	21 (52.5)	30 (62.3)	17 (58.6)	21 (75.0)	0.551
Alcohol consumption, 2–3 times/week N (%)	23 (39.7)	19 (35.2)	20 (32.8)	20 (33.9)	13 (31.7)	20 (41.7)	11 (36.7)	9 (32.1)	0.534
Non-smoking N (%)	44 (74.6)	43 79.6)	48 (77.4)	50 (84.7)	36 (85.7)	35 (74.5)	25 (83.3)	22 (81.5)	0.761
Physical activity, 2–3 times/week N (%)	18 (30.5)	20 (37.0)	15 (24.2)	21 (35.6)	20 (46.5)	15 (31.3)	17 (56.7)	8 (28.6)	0.235
Stress intensity (VAS) ± SD	6.4 ± 2.3	6.1 ± 2.1	6.4 ± 2.7	6.0 ± 1.6	4.6 ± 2.0	5.2 ± 2.1	4.4 ± 2.5	4.8 ± 2.3	<0.001 ^a^
Stress management (VAS) ± SD	6.5 ± 1.6	5.5 ± 1.7	6.4 ± 2.4	5.0 ± 1.6	6.7 ± 2.2	6.7 ± 1.6	6.6 ± 2.3	6.5 ± 2.4	<0.001 ^a^

^a^—ANOVA, Tukey’s honestly significant difference test. Abbreviations: 6ABT—1-week ambulatory BT complex treatment, 11ABT—2 weeks of ambulatory BT complex treatment, 11ABTNT—2 weeks of ambulatory BT complex plus nature therapy treatment, 11SBT—2 weeks of inpatient BT treatment, 11NT—2-weeks nature therapy, 11C—2 weeks control group.

**Table 2 brainsci-15-00165-t002:** Changes in distress symptoms on the GSDS in winter-season study groups after the treatment period.

		6ABT ^a^ N = 59		11ABT ^b^ N = 63		11ABTNT ^c^ N = 63		11SBT ^d^ N = 61		11NT ^e^ N = 43		11C ^f^ N = 51	
		Mean (±SD)	P Effect	Mean (±SD)	P Effect	Mean (±SD)	P Effect	Mean (±SD)	P Effect	Mean (±SD)	P Effect	Mean (±SD)	P Effect
Fatigue	BT	5.4 ± 2.6	<0.001	5.4 ± 2.3	<0.001	4.7 ± 2.7	<0.001	5 ± 2.2	<0.001	4 ± 2.9	0.14	4.8 ± 2.4	0.010
	AT	1.6 ± 2.1	1.2	2.6 ± 2.6	1.2	2.5 ± 2.2	0.8	1.2 ± 1.9	1.5	3.4 ± 2.4	0.2	3.5 ± 2.3	0.1
Sleep	BT	3.1 ± 3.4	<0.001	2.9 ± 2.6	<0.001	4.1 ± 3.6	<0.001	4.5 ± 2.5	<0.001	2.3 ± 2.5	0.95	3 ± 2.4	0.659
	AT	1.5 ± 2.5	0.6	1.8 ± 2.6	0.5	1.7 ± 2.6	0.6	2.7 ± 2.4	0.7	2.3 ± 2.6	0.01	2.8 ± 2.6	−0.2
Pain	BT	2.1 ± 2.6	<0.001	2.4 ± 2.8	<0.001	1.5 ± 2.1	0.731	2.0 ± 3.1	<0.001	2.3 ± 3	0.004	2 ± 2.3	0.703
	AT	0.6 ± 1.1	0.6	0.9 ± 1.5	0.6	1.4 ± 2.2	0.4	1.3 ± 2.1	0.8	1.7 ± 2.4	0.5	2.1 ± 2.4	−0.3
Headache	BT	2.6 ± 2.8	<0.001	2 ± 2.3	<0.001	1.3 ± 1.5	0.067	1.3 ± 1.8	0.003	1.9 ± 2.5	0.014	0.9 ± 1.6	0.857
	AT	0.8 ± 1.5	0.9	0.9 ± 1.7	0.6	0.8 ± 1.7	0.2	0.7 ± 1.4	0.4	0.9 ± 1.7	0.4	1 ± 1.8	−0.3
Anxiety	BT	3.8 ± 3.5	<0.001	4.1 ± 3.2	<0.001	4.4 ± 3.3	<0.001	4.3 ± 2.5	<0.001	2.6 ± 2.6	0.109	2.9 ± 2.9	0.081
	AT	1.6 ± 2.2	0.8	1.9 ± 2.3	1.0	2.3 ± 2.4	0.8	1.9 ± 2.1	1.1	2 ± 2.1	0.3	2.2 ± 2.4	−0.03
Depression	BT	2.1 ± 3.4	<0.001	1.6 ± 2.9	<0.001	2.5 ± 3.5	0.002	1.3 ± 1.7	0.012	0.7 ± 1.5	0.132	0.7 ± 1.6	0.488
	AT	0.8 ± 1.8	0.6	0.6 ± 1.4	0.5	1.4 ± 2.2	0.4	0.6 ± 1.6	0.3	0.4 ± 0.7	0.2	0.5 ± 1.3	−0.2
Memory	BT	3.4 ± 3.2	<0.001	2.9 ± 3.1	<0.001	3.7 ± 3.1	<0.001	3.5 ± 2.4	<0.001	2 ± 2.5	0.110	2.3 ± 2.6	0.525
	AT	0.6 ± 1.3	0.8	1.6 ± 2.2	0.6	1 ± 1.5	0.8	1.9 ± 2	0.6	1.4 ± 2	0.3	2.1 ± 2.6	−0.2
Appetite	BT	1.4 ± 2.6	<0.001	1.3 ± 2.6	0.002	2 ± 3.1	<0.001	1.3 ± 2.2	<0.001	0.1 ± 0.8	0.125	0.4 ± 1.1	0.462
	AT	0.3 ± 1.1	0.5	0.2 ± 0.8	0.4	0.4 ± 1.4	0.5	0.7 ± 1.5	0.6	0.4 ± 1.1	−0.2	0.3 ± 0.8	−0.2
Nausea	BT	0.2 ± 0.9	0.062	0.2 ± 0.6	0.019	0.7 ± 1.4	<0.001	0.2 ± 0.4	<0.001	0.1 ± 0.8	0.156	0.2 ± 0.5	0.351
	AT	0.1 ± 0.3	0.3	0.1 ± 0.3	0.3	0.2 ± 0.8	0.5	0 ± 0.2	0.5	0.5 ± 1.7	−0.2	0.1 ± 0.4	0.1
Vomiting	BT	0.1 ± 0.7	0.340	0.1 ± 0.3	-	0.2 ± 0.7	0.033	0 ± 0.1	0.321	0	0.323	0.1 ± 0.7	0.180
	AT	0.02 ± 0.1	0.1	0.1 ± 0.3	-	0	0.3	0	0.1	0.1 ± 0.3	−0.2	0	0.2
Obstipation	BT	1.6 ± 2.4	<0.001	1.7 ± 2.6	0.005	1.6 ± 2.6	0.002	0.6 ± 1.2	0.005	0.6 ± 1.7	0.246	1.7 ± 2.7	0.007
	AT	0.2 ± 0.6	0.6	0.8 ± 1.4	0.4	0.5 ± 1.3	0.4	0.3 ± 1.2	0.4	0.9 ± 2.3	−0.2	2.4 ± 3.4	−0.4
Diarrhea	BT	0.4 ± 1.3	0.016	0.5 ± 1.3	0.162	0.8 ± 1.8	0.002	0.7 ± 1.8	0.006	0	0.046	0.1 ± 0.2	0.322
	AT	0.0 ± 0.1	0.3	0.6 ± 1.4	−0.2	0.1 ± 0.3	0.4	0.3 ± 0.8	0.4	0.6 ± 1.8	−0.3	0.2 ± 0.8	−0.1
Tingling	BT	0.7 ± 1.7	0.197	2.7 ± 3.2	<0.001	1.6 ± 2.6	0.013	1.5 ± 2	<0.001	0.9 ± 1.9	0.242	1.4 ± 2	0.016
	AT	0.4 ± 1.2	0.2	1.2 ± 2.2	0.5	0.7 ± 1.8	0.3	0.4 ± 1.1	0.8	0.6 ± 1.3	0.2	0.8 ± 1.4	0.4
Rash	BT	1.9 ± 3.0	<0.001	1.6 ± 2.8	0.004	2.3 ± 2.9	0.008	0.8 ± 1.5	0.059	0.8 ± 2.1	0.199	1.1 ± 2.1	0.355
	AT	0.5 ± 1.3	0.5	0.6 ± 1.2	0.4	1.2 ± 2.1	0.4	0.4 ± 1.3	−0.0	1.3 ± 3	−0.2	0.8 ± 1.8	0.1

Abbreviations: 6ABT—1-week of ambulatory BT complex treatment, 11ABT—2 weeks of ambulatory BT complex treatment, 11ABTNT—2 weeks of ambulatory BT complex plus nature therapy treatment, 11SBT—2 weeks of inpatient BT complex treatment in resort, 11NT—2-weeks of nature therapy, 11C—2 weeks control group without treatment, SD—standard deviation; ^a–f^ groups names for the between-group comparison; BT—before treatment, AT—after treatment, effect–effect size calculated as Cohen’s *d*.

**Table 3 brainsci-15-00165-t003:** Changes in distress symptoms on the GSDS in summer season study groups after the treatment period.

		6ABTS ^g^ (N = 30)		11ABTS ^h^ (N = 29)		Between-Group Comparison	
		Mean (±SD)	P Effect	Mean (±SD)	P Effect	BT *p*/sig. Diff. Groups	AT *p*/sig. Diff. Groups
Fatigue	BT	3.5 ± 2.3	0.002	4 ± 2.7	<0.001	0.003	<0.001
	AT	2.1 ± 2.4	0.6	1.5 ± 2.1	1.0	^a^ vs. ^g^, ^b^ vs. ^g^	^a^ vs. ^e^, ^a^ vs. ^f^, ^a^ vs. ^b^, ^c^ vs. ^d^, ^d^ vs. ^e^, ^d^ vs. ^f^, ^d^ vs. ^e^, ^f^ vs. ^h^
Sleep	BT	1.6 ± 2.5	0.718	1.9 ± 2.3	0.039	<0.001	0.002
	AT	1.4 ± 2.2	0.1	0.8 ± 1.6	0.4	^c^ vs. ^e^, ^c^ vs. ^g^, ^c^ vs. ^h^, ^d^ vs. ^e^, ^d^ vs. ^g^, ^d^ vs. ^h^	^d^ vs. ^h^, ^f^ vs. ^h^
Pain	BT	1.1 ± 1.5	0.389	1.9 ± 2.5	0.003	0.033	0.002
	AT	0.8 ± 1.6	0.2	0.8 ± 1.6	0.6	^c^ vs. ^d^, ^d^ vs. ^g^	^a^ vs. ^f^, ^b^ vs. ^f^
Headache	BT	0.9 ± 1.7	0.255	0.9 ± 1.6	0.129	<0.001	0.796
	AT	0.5 ± 1	0.2	0.4 ± 0.9	0.3	^a^ vs. ^c^, ^a^ vs. ^d^, ^a^ vs. ^f^, ^a^ vs. ^g^, ^a^ vs. ^h^	-
Anxiety	BT	1 ± 1.7	0.037	2.6 ± 2.7	<0.001	<0.001	0.001
	AT	0.4 ± 0.8	0.4	0.8 ± 1.5	0.7	^a^ vs. ^g^, ^b^ vs. ^g^, ^c^ vs. ^g^, ^d^ vs. ^g^, ^c^ vs. ^e^	^b^ vs. ^g^, ^c^ vs. ^g^, ^d^ vs. ^g^, ^e^ vs. ^g^, ^f^ vs. ^g^
Depression	BT	0.1 ± 0.7	0.476	0.6 ± 1.4	0.275	<0.001	0.002
	AT	0.0 ± 0.2	0.1	0.3 ± 1.5	0.2	^a^ vs. ^g^, ^c^ vs. ^g^, ^c^ vs. ^e^, ^c^ vs. ^f^, ^c^ vs. ^h^	^c^ vs. ^e^, ^c^ vs. ^g^, ^c^ vs. ^h^
Memory	BT	1.1 ± 1.8	0.637	2.2 ± 2.6	0.003	<0.001	<0.001
	AT	1 ± 1.5	0.1	0.8 ± 1.4	0.6	^a^ vs. ^g^, ^c^ vs. ^g^, ^d^ vs. ^g^, ^c^ vs. ^e^	^a^ vs. ^d^, ^a^ vs. ^f^, ^c^ vs. ^f^
Appetite	BT	0.0 ± 0.2	-	0.2 ± 1	0.742	<0.001	0.200
	AT	0.0 ± 0.2	-	0.3 ± 1.3	−0.1	^c^ vs. ^e^, ^c^ vs. ^f^, ^c^ vs. ^g^, ^c^ vs. ^h^	-
Nausea	BT	0.1 ± 0.3	1	0	-	<0.001	0.015
	AT	0.1 ± 0.4	0	0	-	^a^ vs. ^c^, ^b^ vs. ^c^, ^c^ vs. ^d^, ^c^ vs. ^e^, ^c^ vs. ^f^, ^c^ vs. ^g^, ^c^ vs. ^h^	^a^ vs. ^e^, ^b^ vs. ^e^, ^d^ vs. ^e^, ^e^ vs. ^h^
Vomiting	BT	0.0 ± 0.2	0.326	0	-	0.345	0.146
	AT	0	0.2	0	-	-	-
Obstipation	BT	0.8 ± 2.1	0.083	0.3 ± 1.1	0.326	0.003	<0.001
	AT	0.6 ± 2	0.3	0.2 ± 0.6	0.2	^a^ vs. ^h^, ^b^ vs. ^h^, ^c^ vs. ^h^	^a^ vs. ^f^, ^b^ vs. ^f^, ^c^ vs. ^f^, ^d^ vs. ^f^, ^e^ vs. ^f^, ^f^ vs. ^g^, ^f^ vs. ^h^
Diarrhea	BT	0.0 ± 0.2	0.326	0.2 ± 0.6	0.477	0.003	0.002
	AT	0.1 ± 0.3	−0.2	0.1 ± 0.4	0.1	^c^ vs. ^e^, ^c^ vs. ^f^	^a^ vs. ^b^, ^b^ vs. ^c^
Tingling	BT	0.5 ± 0.9	0.095	1.1 ± 1.9	0.035	<0.001	0.015
	AT	0.2 ± 0.5	0.3	0.3 ± 1	0.4	^a^ vs. ^b^, ^b^ vs. ^e^, ^b^ vs. ^g^, ^b^ vs. ^h^	^b^ vs. ^d^,^b^ vs. ^g^
Rash	BT	0.9 ± 2.1	0.03	0.7 ± 1.5	0.313	0.004	0.034
	AT	0.5 ± 1.4	0.4	0.3 ± 1	0.2	^c^ vs. ^d^, ^c^ vs. ^h^	^c^ vs. ^d^, ^c^ vs. ^h^, ^e^ vs. ^d^, ^e^ vs. ^h^

Abbreviations: 6ABTS—1 week of ambulatory BT complex treatment in summer, 11ABTS—2 weeks of ambulatory BT complex treatment in summer; ^a^—6ABT group, ^b^—11ABT group, ^c^—11ABTNT group, ^d^—11SBT group, ^e^—11NT group, ^f^—11C group, ^g^—6ABTS group, ^h^—11ABTS group, SD—standard deviation; ^a–h^—groups names for the between-group comparison; BT—before treatment, AT—after treatment, vs.—versus, effect—effect size calculated as Cohen’s *d*. In-between group comparison made using POSTHOC = TUKEY T2, ANOVA.

**Table 4 brainsci-15-00165-t004:** The findings from the regression analysis on distress intensity.

	Unstandardized Coefficients		Standardized Coefficients	t	Sig.	95.0% Confidence Interval for B		Collinearity Statistics	
	B	Std. Error	Beta			Lower Bound	Upper Bound	Tolerance	VIF
(Constant)	40.40	12.17		3.32	0.001	16.42	64.37		
Work and social adaptation (WSAS)	0.46	0.10	0.32	4.78	<0.001	0.27	0.64	0.65	1.53
Perceived stress scale (PSS-10)	0.26	0.20	0.11	1.34	0.181	−0.123	0.647	0.39	2.54
Fatigue (FAS)	−0.16	0.21	−0.05	−0.77	0.44	−0.57	0.25	0.58	1.74
Anxiety_state (STAIS-5)	1.97	0.47	0.33	4.16	<0.001	1.036	2.898	0.47	2.15
Anxiety_trait (STAIT-5)	−1.27	0.38	−0.29	−3.37	<0.001	−2.01	−0.53	0.39	2.55
Depression (CESD-R)	−0.15	0.12	−0.11	−1.28	0.20	−0.38	0.08	0.38	2.65
Integrative outcomes scale (AIOS)	−2.06	0.57	−0.26	−3.60	<0.001	−3.19	−0.93	0.56	1.79
Sleep (single item SQS)	−0.31	0.49	−0.04	−0.64	0.53	−1.27	0.65	0.65	1.55
Systolic blood pressure, mmHg	0.05	0.07	0.05	0.69	0.49	−0.09	0.20	0.49	2.03
Diastolic blood pressure, mmHg	−0.17	0.12	−0.12	−1.47	0.14	−0.40	0.06	0.47	2.15
Heart rate, beats/min	0.04	0.07	0.03	0.52	0.61	−0.11	0.19	0.88	1.14
Quality of life (5-point Likert scale)	0.79	1.49	0.04	0.53	0.60	−2.15	3.72	0.54	1.87
Pain, VAS	0.71	0.43	0.10	1.66	0.10	−0.13	1.55	0.79	1.27
Saliva cortisol, nmol/L	0.09	0.41	0.01	0.22	0.83	−0.72	0.90	0.93	1.07

**Table 5 brainsci-15-00165-t005:** Stress intensity before and after treatment by gender across winter intervention groups.

Study Groups		6ABT				11ABT				11ABTNT				11SBT				11NT				11C			
Parameters		Mean (SD)	Mean Diff	Effect Size		Mean (SD)	Mean Diff	Effect Size		Mean (SD)	Mean Diff	Effect Size		Mean (SD)	Mean Diff	Effect Size		Mean	Mean Diff	Effect Size		Mean (SD)	Mean Diff	Effect Size	
					*t*, df, *p*				*t*, df, *p*				*t*, df, *p*				*t*, df, *p*				*t*, df, *p*				*t*, df, *p*
Stress intensity BT	Male	5.08 (2.33)	−1.63	0.719	*t* = −2.329; df = 56; *p* = 0.024	5.43 (1.99)	0.614	-	*t* = *0*,721; df = 52; *p*= 0.479	6.45 (3.12)	0.035	-	*t* = 0.045; df = 30.03; *p* = 0.965	6.18 (1.81)	0.245	-	*t* = −0.521; df = 59, *p* = 0.604	3.50 (2.07)	−1.235	-	*t* = −1.413; df = 38, *p* = 0.166	5.33 (1.56)	0.133	-	*t* = −0.194; df = 45 *p* = 0.847
	Female	6.71 (2.20)				6.04 (2.12)				6.41 (2.37)				5.93 (1.58)				4.74 (1.96)				5.20 (2.19			
Stress intensity AT	Male	1.69 (0.86)	1.40	0.794	*t* = −3.29; df = 52.36; *p* = 0.002	3.29 (1.98)	0.193	-	*t* = *0.233*; df = 52; *p* = 0.816	3.70 (2.60)	0.432	-	*t* = 0,658; df = 29.80; *p* = 0.515	3.76 (1.39)	0.469	-	*t* = 1.151; df = 59, *p* = 0.254	3.00 (1.53)	−1.235	-	*t* = −1.548; df = 39, *p* = 0.130	4.09 (1.97)	−0.081	-	*t* = −0.115; df = 44, *p* = 0.909
	Female	3.09 (2.34)				3.48 (2.04)				3.27 (1.95)				3.30 (1.44)				4.24 (1.97)				4.17 (2.04)			

Abbreviations: 6ABT—1-week of ambulatory BT complex treatment, 11ABT—2 weeks of ambulatory BT complex treatment, 11ABTNT—2 weeks of ambulatory BT complex plus nature therapy treatment, 11SBT—2 weeks of inpatient BT complex treatment, 11NT—2-weeks of nature therapy, 11C—2 weeks control group, SD—standard deviation; BT—before treatment, AT—after treatment, effect—effect size calculated as Cohen’s *d*.

**Table 6 brainsci-15-00165-t006:** Stress intensity before and after treatment by gender across summer intervention groups.

Study Groups		6ABTS				11ABTS			
Parameters		Mean (SD)	Mean Diff	Effect Size		Mean (SD)	Mean Diff	Effect Size	
					*t*, df, *p*				*t*, df, *p*
Stress intensity BT	Male	4.14 (2.67)	−0.18	-	*t* = −0.177; df = 36; *p* = 0.860	4.75 (2.44)	−0.287	-	*t* = −0.305; df = 33; *p* = 0.762
	Female	4.32 (2.46)				5.04 (2.31)			
Stress intensity AT	Male	4.25 (2.18)	0.90	-	*t* = 0.839; df = 28; *p* = 0.409	4.71 (2.36)	2.289	0.966	*t* = 2.975; df = 26; *p* = 0.006
	Female	3.35 (1.98)				2.43 (1.53)			

Abbreviations: 6ABTS—1 week of ambulatory BT complex treatment in summer, and 11ABTS—2 weeks of ambulatory BT complex treatment in summer; SD—standard deviation; BT—before treatment, AT—after treatment, effect—effect size calculated as Cohen’s *d*.

**Table 7 brainsci-15-00165-t007:** Salivary cortisol before and after treatment by gender across study groups.

Study Groups		6ABT				11ABT				11ABTNT				11BTS				11NT				11C			
Parameters		Mean (SD)	Mean Diff	Effect Size		Mean (SD)	Mean Diff	Effect Size		Mean (SD)	Mean Diff	Effect Size		Mean (SD)	Mean Diff	Effect Size		Mean	Mean Diff	Effect Size		Mean (SD)	Mean Diff	Effect Size	
					*t*, df, *p*				*t*, df, *p*				*t*, df, *p*				*t*, df, *p*				*t*, df, *p*				*t*, df, *p*
C-BT	Male	4.81 (3.29)	2.07	0.784	*t* = 2.179; df = 14.06; *p* = 0.047	3.50 (2.27)	0.694	-	*t* = 0.989; df = 42; *p* = 0.329	4.59 (2.87)	1.258	-	*t* = 1.621; df = 57; *p* = 0.110	2.94 (1.43)	−0.592	-	*t* = −0.986; df = 58, *p* = 0.328	2.94 (1.28)	0.424	-	*t* = 0.675; df = 26; *p* = 0.506	1.97 (0.80)	−0.698	-	*t* = −0.811; df = 44, *p* = 0.422
	Female	2.74 (1.77)				2.80 (1.46)				3.33 (2.75)				3.53 (2.30)				2.52 (1.27)				2.67 (2.79			
C-AT	Male	3.00 (1.79)	0.62	-	*t* = 1.175; df = 56; *p* = 0.245	2.26 (1.43)	0.240	-	*t* = 0.380; df = 42; *p* = 0.706	3.63 (2.20)	1.064	0.547	*t* = 2.071; df = 57; *p* = 0.043	2.26 (1.64)	−0.656	-	*t* = −1.149; df = 58, *p* = 0.255	2.55 (1.40)	−0.422	-	*t* = 0.567; df = 26; *p* = 0.575	2.25 (1.41)	−0.482	-	*t* = −0.715; df = 44, *p* = 0.478
	Female	2.38 (1.64)				2.01 (1.44)				2.57 (1.66)				2.92 (2.13)				2.13 (1.53)				2.74 (2.08)			

Abbreviations: C—saliva cortisol, 6ABT—1-week of ambulatory BT complex treatment, 11ABT—2 weeks of ambulatory BT complex treatment, 11ABTNT—2 weeks of ambulatory BT complex plus nature therapy treatment, 11BTS—2 weeks of inpatient BT complex treatment in resort, 11NT—2-weeks of nature therapy, 11C—2 weeks control group without treatment, SD—standard deviation; BT—before treatment, AT—after treatment, effect—effect size calculated as Cohen’s *d*.

## Data Availability

The raw data that supports the conclusions of this article will be made available by the authors upon reasonable request.

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
