# Peer review of "Balneotherapy as a Complementary Intervention for Stress and Cortisol Reduction: Findings from a Randomized Controlled Trial"

_brainsci, 2025, doi:10.3390/brainsci15020165_

Round 1
Reviewer 1 Report
Comments and Suggestions for Authors
The investigation is based on a randomized controlled trial scientifically well designed, with acceptable ethical issues (please indicate if the consent was a written consent).
The results are in accordance with previous knowledge based on less robust scientific data, making the study particularly useful.
The paper is well-written. No remarks must be made about methodology, results, discussion, conclusion, bibliography.
Accept for publication after the written consent modified.
Author Response
The investigation is based on a randomized controlled trial scientifically well designed, with acceptable ethical issues (please indicate if the consent was a written consent).
Thank you for your comments! The necessary corrections have been made. Regarding your query about consent, it was indeed a written consent obtained from all participants involved in the study. Please let me know if you need any further clarification. Wishing you all the best!
Reviewer 2 Report
Comments and Suggestions for Authors
Regarding the manuscript brainsci-3412916 „Balneotherapy as a Complementary Intervention for Stress and 2 Cortisol Reduction: findings from a randomized controlled trial”, I communicate the following: It’s a valuable article because it contains important information about management and new therapies for reducing stress, enhance mental health, and promote overall well-being. Also, this study aimed to evaluate the seasonal effects of balneotherapy on distress, salivary cortisol, and integrative outcomes, and provides robust evidence for developing holistic, seasonally optimized strategies to aid stress management and promote neurological health.
1. The abstract. The abstract is concise and well-written, effectively summarizing the key aspects of the manuscript. It clearly outlines the background, objectives, methodology, and key findings, providing a strong clinical study. The results are presented with sufficient detail to highlight the significant advantages of balneotherapy on stress management. The conclusion is clear and aligns well with the manuscript’s overall findings.
2. Keywords are significant.
3. The introduction. The manuscript addresses the comparative efficacy of balneotherapy The topic is well-chosen and highly relevant, with significant implications for stress management, given seasonally optimized strategies for balneal treatment.
4. Structure. The manuscript is well-structured, following a clear format: introduction, methodology, results, discussion, and conclusions. Each section contributes effectively to building a comprehensive understanding of the topic.
5. Material and methods. The authors employed a rigorous methodology for study selection, adhering to CONSORT guidelines. The detailed use of the SPSS program is commendable and adds the robustness of the analysis. Also, the large number of subjects, divided into 6 groups, confers a rigorous scientific rigor.
6. Results. The results are clearly presented, emphasizing the advantages of the analyzed therapies. The data are well-synthesized, providing a comprehensive overview of the impact of these interventions on stress and cortisol levels.
7. Discussion. The discussion is well-balanced, acknowledging the included studies' strengths and limitations. The authors effectively highlight the practical implications of their findings and provide a foundation for future research.
8. The conclusion. The conclusions are well-supported by the presented data and highlight the significant stress-reducing benefits of combining natural resources, particularly balneotherapy and nature therapy,
9. References. The manuscript adheres to high scientific standards. The references are relevant and reflect meticulous research.
10. Language and style. The language and style of the manuscript are professional, precise, and appropriate for a scientific journal. The authors demonstrate a clear and thorough command of academic writing, with well-structured sentences and logical flow throughout the text. Technical terminology is used accurately and consistently, reflecting the authors' expertise in the field.
Final recommendation
I recommend the manuscript for publication, as it makes a significant contribution to the field of natural resources, particularly balneotherapy and nature therapy,
Author Response
Thank you very much for the excellent evaluation of the article and recommendation. We will continue to work in this field, wishing you all the best!
Reviewer 3 Report
Comments and Suggestions for Authors
The paper by Rapolienė et al. is devoted to studying the seasonal effect of balneotherapy on stress symptoms and salivary cortisol level. A randomized controlled, single-blinded parallel-group trial was conducted in six medical spa centers in Lithuania (Informed consent for participation was obtained from all subjects involved in the study). The authors demonstrated the stress-reducing effects of balneotherapy as a reduction in salivary cortisol level, an increase in well-being and an improved stress management. The efficacy was enhanced during winter interventions. The paper is well structured, the 4 tables and 4 figures complement the text well. Abstract gives all the necessary information about the contents of the paper, keywords are appropriately chosen. The reference list covers the relevant literature adequately (the authors cite 39 sources).
However, I have some remarks:
1. It would be better if the authors emphasized what exactly is new about their study, since by now a lot of data has been accumulated on the effects of balneotherapy on stress indicators (including cortisol levels).
2. Also, in my opinion, the aim of the study should be written in more detail, this would help to interest a larger number of readers. The current formulation of the aim is too general. It could be stated that it is planned to study not just "distress" and "integrative outcomes", but "distress symptoms using GSDS scale (General Symptoms Distress Scale)" and "well-being using AIOS scale (Arizona Integrative Outcomes Scale)".
3. The authors do not comment on whether any gender differences were found in this study. This could be an interesting aspect, since stress is known to have a strong effect on hormonal status. If the sample sizes did not allow the authors to draw conclusions about the influence of gender on the results of the study, then perhaps it is worth mentioning this in the Limitations and future directions section.
Author Response
Comment 1. It would be better if the authors emphasized what exactly is new about their study, since by now a lot of data has been accumulated on the effects of balneotherapy on stress indicators (including cortisol levels).
Thank you for your insightful question! We agree with your remark and have corrected the introduction to better highlight the study's novelty. In addition, while many studies have explored balneotherapy and its effects on stress, our study makes several unique contributions: We assess both objective (cortisol levels) and subjective stress (psychological questionnaires), providing a more comprehensive evaluation; We focus on specific stress-related symptoms like sleep disturbances, fatigue, and anxiety, rather than general stress outcomes; This is the first study to examine seasonal variations in BT’s effectiveness on stress and wellness; according your suggestion we also test different BT modalities and explore gender differences.
Comment 2. Also, in my opinion, the aim of the study should be written in more detail, this would help to interest a larger number of readers. The current formulation of the aim is too general. It could be stated that it is planned to study not just "distress" and "integrative outcomes", but "distress symptoms using GSDS scale (General Symptoms Distress Scale)" and "well-being using AIOS scale (Arizona Integrative Outcomes Scale)".
Thank you for your comment. We have revised the study aim description to provide more detail, as you suggested.
Comment 3. The authors do not comment on whether any gender differences were found in this study. This could be an interesting aspect, since stress is known to have a strong effect on hormonal status. If the sample sizes did not allow the authors to draw conclusions about the influence of gender on the results of the study, then perhaps it is worth mentioning this in the Limitations and future directions section.
Thank you for your evaluation and helpful comments. I have carefully considered all of them, added more information, and updated the calculations. The revisions are highlighted in green, including the aim, the introduction section on novelty, and the results and discussion related to gender differences in stress. Thank you once again for your valuable contribution to the article. Wishing you all the best!